# Discovery of Novel Pimprinine and Streptochlorin Derivatives as Potential Antifungal Agents

**DOI:** 10.3390/md20120740

**Published:** 2022-11-25

**Authors:** Jing-Rui Liu, Jia-Mu Liu, Ya Gao, Zhan Shi, Ke-Rui Nie, Dale Guo, Fang Deng, Hai-Feng Zhang, Abdallah S. Ali, Ming-Zhi Zhang, Wei-Hua Zhang, Yu-Cheng Gu

**Affiliations:** 1Jiangsu Key Laboratory of Pesticide Science, College of Sciences, Nanjing Agricultural University, Nanjing 210095, China; 2State Key Laboratory Breeding Base of Systematic Research Development and Utilization of Chinese Medicine Resources, School of Pharmacy, Chengdu University of Traditional Chinese Medicine, Chengdu 611137, China; 3Key Laboratory of Integrated Management of Crop Diseases and Pests, Ministry of Education, Department of Plant Pathology, College of Plant Protection, Nanjing Agricultural University, Nanjing 210095, China; 4Department of Microbiology, Faculty of Agriculture, Cairo University, Giza 12613, Egypt; 5Syngenta Jealott’s Hill International Research Centre, Bracknell RG42 6EY, Berkshire, UK

**Keywords:** pimprinine, streptochlorin, synthesis, antifungal activity, EC_50_, SAR, molecular docking

## Abstract

Pimprinine and streptochlorin are indole alkaloids derived from marine or soil microorganisms. In our previous study, they were promising lead compounds due to their potent bioactivity in preventing many phytopathogens, but further structural modifications are required to improve their antifungal activity. In this study, pimprinine and streptochlorin were used as parent structures with the combination strategy of their structural features. Three series of target compounds were designed and synthesized. Subsequent evaluation for antifungal activity against six common phytopathogenic fungi showed that some of thee compounds possessed excellent effects, and this is highlighted by compounds **4a** and **5a**, displaying 99.9% growth inhibition against *Gibberella zeae* and *Alternaria Leaf Spot* under 50 μg/mL, respectively. EC_50_ values indicated that compounds **4a**, **5a**, **8c,** and **8d** were even more active than Azoxystrobin and Boscalid. SAR analysis revealed the relationship between 5-(3′-indolyl)oxazole scaffold and antifungal activity, which provides useful insight into the development of new target molecules. Molecular docking models indicate that compound 4a binds with leucyl-tRNA synthetase in a similar mode as AN2690, offering a perspective on the mode of action for the study of its antifungal activity. These results suggest that compounds **4a** and **5a** could be regarded as novel and promising antifungal agents against phytopathogens due to their valuable potency.

## 1. Introduction

Natural products, small molecules isolated from biological sources, play a highly significant role in medicine and agrochemical innovation, and the repertoire of natural products offers tremendous opportunities for chemical biology and drug discovery [1,2,3,4]. Approximately two-thirds of all approved small-molecule drugs from January 1981 to September 2019 owe their origins to natural products [5]. Pimprinine is an indole alkaloid produced by many species of *Streptomyces*, first isolated from the filtrates of *Streptomyces pimprina* cultures in 1963 [6]. For decades, the marine environment has also provided a source of novel bioactive and structurally diverse natural products [7]. Streptochlorin, with a structure similar to that of pimprinine, is a bacterial metabolite originally isolated from marine *Streptomyces* sp. By H. Watanabe in 1988 [8]; its structure is shown in Figure 1. Members of this natural 5-(3′-indolyl)oxazole family (Figure 1), including pimprinethine, pimprinaphine, WS-30581 A and B, labradorins 1 and 2, pimprinol A, B, and C, martefragin A, deaminomartefragin A, almazole C and D, breitfussin A and B, and dipimprinine E and F, exhibit a wide range of potent biological activity, such as antioxidation [9,10], anticancer [11,12], antiviral [13,14], anti-angiogenesis [15], and antibiotic properties [16], anti-cell proliferation, [17] and pesticidal activity [18]. Pimprinine and streptochlorin were promising antifungal substances due to their good bioactivity in preventing many phytopathogens in our previous study [19]. Bio-screening conducted by Syngenta showed that streptochlorin displayed excellent antifungal activity against *Pythium dissimile*, *Botrytis cinerea*, *Zymoseptoria tritici*, *Pyriculariaory zae*, *Fusarium culmorum,* and *Rhizoctonia solani* in artificial media [20]. Meanwhile, pimprinine and streptochlorin lack potency under lower concentrations and are rarely extended to the next stage of study, as they are not potent enough to be used as antifungal agents, and the mode of action for their antifungal activity is still unclear. In our latest study on the structural optimizations including different modifications at the indole ring and oxazole ring [21,22,23,24,25,26], we found that the 5-(3′-indolyl)oxazole core was the essential moiety for maintaining antifungal activity.

In this study, as a continuation of our extensive research program to discover novel bioactive lead compounds, pimprinine and streptochlorin were used as the parent structures to carry out structural optimization (Figure 2), with the structural features combination strategy of these two indole alkaloids. Three series of target compounds were designed and synthesized, aiming to discover synthetic derivatives with a modified chemical structure and improved antifungal activity. The structure–activity relationships (SAR) around pimprinine and streptochlorin were also analyzed, and the molecular docking of streptochlorin with a potential target enzyme was further performed.

## 2. Results and Discussion

### 2.1. Synthetic Chemistry

Novel pimprinine and streptochlorin derivatives were synthesized as depicted in Figure 1 and Figure 2. In this approach, we described a synthesis of 5-(3′-indolyl)oxazoles alkaloids in one pot with the reported method [27], employing 3-actylindole and amino acids as substrates to be transformed into natural products. In this reaction process, we used indole as the starting material. After the acylation of indole, which gives 3-actylindole, the common precursor indole α-keto aldehyde, which was generated by an iodination/Kornblum oxidation sequence from 3-actylindole, was trapped in situ by an amino acid via a condensation/decarboxylation/annulation/oxidation reaction sequence, to eventually approach the natural products. As reported in the literature, two equivalents of I_2_ were used, one equivalent of I_2_ as a halogenation reagent and the other equivalent as the oxidation reagent. In our modified synthetic process, we optimized the addition time of the reaction: 1.1 equivalents of I_2_ were used in the initial iodination reaction, and the rest of the 0.9 equivalent I_2_ was added after the addition of amino acid. This improved method can increase the yield by 10%. Compound data, Copies of the NMR spectra, and HR-MS (ESI) spectra can be downloaded at Appendix A.

Therefore, we accomplished the synthesis of 5-(3-indolyl)oxazoles alkaloids (Table 1, Table 2 and Table 3), including pimprinine, pimprinethine, and labradorins 1, as well as their derivatives, directly. The subsequent NCS or NBS halogenation yielded novel 4-chloro-5-(3-indolyl)oxazoles and 4-bromo-5-(3-indolyl)oxazoles, respectively, including the marine natural product streptochlorin [11]. Particularly worth mentioning is that this is the first time that streptochlorin has been efficiently synthesized using this three-step method.

### 2.2. Antifungal Activity and Structure–Activity Relationships (SAR)

The antifungal activity of pimprinine, streptochlorin, their derivatives, and the positive controls was evaluated with the mycelium growth rate method against six common phytopathogenic fungi, including *Alternaria Leaf Spot* (ALL), *Alternaria solani* (ALS), *Botrytis cinerea* (BOT), *Colletotrichum lagenarium* (COL), *Gibberella zeae* (GIB), and *Rhizoctorzia solani* (RHI), at a concentration of 50 μg/mL. The screening results are given in Table 4 and Table 5.

It was observed that compounds **3a**, **4a**, **5a**, **8d**, and **8g** showed significant antifungal activity against four kinds of fungi during the primary screening. The antifungal activity ranged from 60.3% to 99.9% at 50 μg/mL, and this was highlighted by the inhibition rates of these four molecules against *Colletotrichum lagenarium*, ranging from 88.3% to 94.6%. The most active compounds, **4a** (streptochlorin) and **5a,** were also compared with commercial fungicides in radar charts shown in Figure 3, and this indicates that **4a** and **5a** showed more effective or equivalent control against the four kinds of fungi than the positive controls.

In order to compare the antifungal activity of the synthesized target compounds with that of the most frequently used commercial fungicides Boscalid, Azoxystrobin, and Carbendazim, EC_50_ values of the highly active compounds (**4a**, **5a**, **8c**, **8d**) were further measured, as these compounds showed equivalent or even better performance than the positive controls. As shown in Table 6, it was noticed that the EC_50_ value of **4a** against *Botrytis cinerea* was as low as 0.3613 μg/mL, which is more effective than Boscalid (5.2606 μg/mL) and Azoxystrobin (4.3516 μg/mL), and compound **5a** exhibited better activity against *Alternaria leaf spot* (3.4086 μg/mL) and *Colletotrichum lagenarium* (8.1215 μg/mL) than their corresponding controls. Moreover, the antifungal activity of **5a** was equivalent to that of Carbendazim and Boscalid against *Gibberella zeae* and *Rhizoctonia solani*, respectively.

In spite of the difficulties in finding clear structure–activity relationships from the biological data, some broad conclusions can still be drawn.

First, the halogenated compounds (compounds **4** and **5**) generally displayed more potent activity and a broader antifungal spectrum in the artificial media assays compared with their unhalogenated counterparts (compounds **3**). On the whole, the compound whose 4-position of the oxazole ring was substituted by a halogen atom (Cl, Br) showed better antifungal activity than those that were not halogenated, though compound **3a** also demonstrated 97.7% and 98.3% inhibition against *Botrytis cinerea* and *Rhizoctonia solani*, respectively. This was equivalent to or even more active than the halogenated counterparts.

Second, bio-screening data of the antifungal activity indicated that the compound with H, Me, or Et substituted at the 2-position of the oxazole ring exhibited more potent antifungal activity than those with other substituents. This is highlighted by compounds **3a**, **3b**, and **3c** and their corresponding halogenated counterparts **4a**, **4b**, and **4c** and **5a**, **5b**, and **5c**, which showed more effective control than the compound with a larger substituent. Therefore, we kept the substituent at the 2-position of oxazole as methyl or ethyl and introduced various substituents at the indole ring, such as methyl and halogen, and these modifications resulted in a number of highly active compounds **(8d**, **8g**, **8k,** and **10d**), some of which showed high inhibitory effects against *Colletotrichum lagenarium*, such as **8d** (91.2%) and **8g** (92.8%).

Third, the synthesized derivatives of pimprinine and streptochlorin seemed more active in inhibiting the growth of *Alternaria leaf* spot, *Colletotrichum lagenarium*, *Gibberella zeae*, and *Rhizoctonia solani*, in particular for *Rhizoctonia solani*, the soilborne pathogen that caused rice sheath blight, resulting in annual severe losses in yield and quality in many rice production areas worldwide. Further, 13 of the 49 target molecules showed growth inhibition above 70%, and this was highlighted by compounds **3a** and **5a**, which displayed 98.3% and 96.1% growth inhibition, respectively—even more active than that of Osthole and Boscalid.

### 2.3. Molecular Modeling

The mode of action for the antifungal activity of pimprinine and streptochlorin derivatives is still not clear, though it has been reported that pimprinine is a potent inhibitor of monoamine oxidase [28,29]. Molecular docking in our previous study indicated that a pimprinine and streptochlorin derivative binds with leucyl-tRNA synthetase in a similar mode as AN2690, offering a perspective on the potential target for the antifungal activity of this series of indole natural products [26].

We performed molecular modeling studies using the X-ray structure of *Thermus thermophiles* LeuRS (PDB ID: 2V0C). The protein was downloaded in high resolution solved at 1.85 Å from https://www.rcsb.org/ (accessed on 13 Jan 2021). The protein crystal structure of *t*LeuRS [30] and the selected ligand **4a** were prepared by Discovery Studio 2.5, and the subsequent docking study was performed using MOE. After the molecular docking, the best binding mode of **4a** (yellow in Figure 4) was selected according to the results of the docking energy, as compared with the AN2690-AMP in the *t*LeuRS (Figure 4).

The simulated models and scores indicated that compound **4a** putatively binds with *t*LeuRS in a similar mode as AN2690. It formed two weak hydrogen bonds with residues Thr248 and Thr252, a C–H**^…^**π interaction with residue Asp344, cation–π interactions with residue Arg346, and a halogen bond between Cl and Arg346 (Figure 4).

## 3. Materials and Methods

### 3.1. Chemistry

All general reagents and substrates commercially available were purchased from Alfa Aesar (Beijing, China) or through Nanjing JG-Chemicals (Nanjing, China) and were used without further purification. All solvents and liquid reagents were dried by standard methods in advance and distilled before use. Column chromatography was performed using silica gel (200–300 mesh). Melting points were determined using a Büchi M-560 melting point apparatus. ^1^H NMR and ^13^C NMR spectra were recorded on a Bruker Avance 400 MHz spectrometer (Rheinstetten, Germany) in a DMSO-*d*_6_, CD_3_OD-*d*_4_ or Acetone-*d*_6_ solution. The chemical shifts (*δ*), multiplicity (s = singlet, d = doublet, t = triplet, m = multiplet, q = quadruple), coupling constants (Hz), and coupling constants (*J*) relative to tetramethylsilane are given in parts per million (ppm) and Hertz (Hz), respectively. HR-MS (ESI) spectra were obtained on an Agilent Technologies 6540 UHD Q-TOF LC-MS (Palo Alto, CA, USA).

Further, 3-acetylindoles (**2** and **7**) was synthesized using the reported methods [31] or purchased through Nanjing Crystal Chemicals Technology Co., Ltd (Nanjing, China). All the reaction yields were not optimized.

#### 3.1.1. Preparation of 1-(1H-indol-3-yl)ethan-1-one (**2**)

Compound **1** (3.51 g, 30.00 mmol) was dissolved in anhydrous CH_2_Cl_2_ (20 mL) and cooled to 0–5 °C under a N_2_ atmosphere. SnCl_4_ (4.2 mL, 36.0 mmol) was added dropwise, then warmed to room temperature and allowed to react for 0.5 h, followed by the dropwise addition of 2.1 mL (30.0 mmol) CH_3_COCl. The mixture was left to react for about another 2 h. When TLC monitoring showed that the reaction was complete, it was quenched with water and extracted with CH_2_Cl_2_ three times (3 × 50 mL). The organic layer was washed with water and brine and dried over anhydrous Na_2_SO_4_. After rotary evaporation, the residue was purified by column chromatography over silica gel (eluent: petroleum ether/acetone = 10:1) to give the pure compound **2**.

#### 3.1.2. Preparation of 2-substituted-5-(1H-indol-3-yl)-oxazole (**3**)

A mixture of compound **2** (0.64 g, 4.0 mmol), I_2_ (0.66 g, 4.4 mmol) in DMSO (3.0 mL), was stirred at 110 °C for 1 h, until almost full conversion of the substrates was indicated by TLC analysis, then α-amino acid (8.0 mmol) and I_2_ (0.54 g, 3.6 mmol) were added, and the mixture was stirred at 110 °C for 10-15 min. Then, 50 mL water and 30 mL saturated brine solution were added to the mixture and extracted with CH_2_Cl_2_ three times (3 × 50 mL). The extract was washed with 10% Na_2_S_2_O_3_ solution (3 × 50 mL), dried over anhydrous Na_2_SO_4_, and concentrated under reduced pressure. The crude product was purified by column chromatography on silica gel (eluent: petroleum ether/acetone = 8:1) to afford the product **3**. Information for the compounds is shown in Table 1.

#### 3.1.3. General Procedure for the Synthesis of 2-substituted-4-halogen-5-(1H-indol-3-yl)oxazole (**4** or **5**)

To a stirred solution of **3** (0.50 mmol) in THF-CCl_4_ (10 mL, 1:1 in *v/v*), NCS or NBS (0.55 mmol) was added, and the resulting mixture was heated at 45 °C for about 8 h, then allowed to cool down. The solvent was removed under reduced pressure, and the crude product was purified by flash column chromatography (eluent: petroleum ether/acetone = 8:1), in order to give the desired intermediate compounds **4** or **5**, respectively. Information for the compounds is shown in Table 2.

#### 3.1.4. Synthesis of Substituted 5-(1H-indol-3-yl)-2-methyloxazoles (**8**–**10**)

The synthetic procedures for compounds **8**–**10** were the same as those described in the general procedure for the synthesis of compounds **3**–**5**. The synthetic route is shown in Figure 2, and information for the compounds is shown in Table 3.

### 3.2. Biological Assays

Antifungal activity testing of the target compounds was carried out using mycelia growth-inhibitory rate methods (Figure 5). The samples were tested at a concentration of 50 μg/mL. Boscalid, Carbendazim, Osthole, and Azoxystrobin were used as positive controls. The tested fungi were provided by the Laboratory of Plant Disease Control, Nanjing Agricultural University, and the experimental procedure for the antifungal activity was performed according to the paper from Department of Plant Pathology, Nanjing Agricultural University [32]. The strains were activated in PDA at 25 °C for 2–15 days to obtain new mycelia, and the edge of the mycelia was punched before the antifungal activity assay. The results of the testing on target compounds against *Alternaria leaf spot, Alternaria solani*, *Botrytis cinerea, Colletotrichum lagenarium, Gibberella zeae,* and *Rhizoctonia solani* are listed in Table 4 and Table 5.

### 3.3. Molecular Modeling Strategy

Discovery Studio 2.5 was used for the preparation of the protein and ligand. We deleted the water molecules in the protein, supplemented incomplete amino acid residues, and hydrotreated the protein. For ligand molecules, we used the software to draw small molecules, optimize the three-dimensional structure, conduct hydrotreatment, and complete energy minimization. The corresponding parameter setting panel was opened, and generally, the default value was set. Then, we clicked ‘Run’ to obtain the processed ligand molecule. Subsequent semi-flexible docking was performed using MOE. The number of placement poses was 50. After the molecular docking, the best binding mode was selected for analysis.

## 4. Conclusions

In conclusion, the natural products pimprinine and streptochlorin were used as the parent structures with the combination strategy of their structural features. Three series of derivatives were effectively synthesized from the starting material indole, using Vilsmeier–Haack acetylation, iodination/Kornblum oxidation, and oxazole annulation in a sequential order. The antifungal activity of 49 designed derivatives against six common phytopathogenic fungi was evaluated at a concentration of 50 μg/mL, and the results showed some of the target molecules possessed excellent antifungal activity, such as compounds **3a**, **4a**, **5a**, **8c**, **8d,** and **8g**, displaying more than 90% growth inhibition against at least one of the tested fungi. The compounds showed antifungal activity equivalent to or even more effective than the positive controls, and this was highlighted by compounds **3a**, **4a,** and **5a**, which displayed over 90% growth inhibition against three kinds of fungi, showing a very broad antifungal spectrum. Especially for the compounds **4a** and **5a**, EC_50_ values against *Botrytis cinerea* were as low as 0.3613, and 1.1283 μg/mL, respectively, which represents better antifungal activity than that of the commonly used fungicides Azoxystrobin and Boscalid. The SAR study revealed the relationship between the 5-(3′-indolyl)oxazole scaffold and antifungal activity, which gives a useful insight into the development of new target molecules. Molecular docking models indicate that **4a** binded with leucyl-tRNA synthetase in a similar mode as AN2690, offering a perspective on the study of the mode of action of the antifungal activity of pimprinine and streptochlorin derivatives. These results therefore suggest that compounds **4a** and **5a** could be regarded as novel and promising antifungal agents against phytopathogens due to their valuable potency.

## Data Availability

The data presented in this study are available in the manuscript and in the Appendix A.

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
