# Peer review of "Discovery of Novel Pimprinine and Streptochlorin Derivatives as Potential Antifungal Agents"

_marinedrugs, 2022, doi:10.3390/md20120740_

Round 1

Reviewer 1 Report

The manuscript entitled “Discovery of novel pimprinine and streptochlorin derivatives as potential antifungal agents” by Jing-Rui Liu et al. describes their latest study on the marine active compounds streptochlorin and its derivatives. As we know, developing new green pesticides is becoming increasingly difficult. Liu and co-authors used pimprinine and streptochlorin as the parent structures, designed and synthesized three series of new derivatives. The antifungal activity testing revealed some compounds showed excellent and broad-spectrum antifungal activity, several compounds were even more active than the most used fungicides. The SAR analysis and molecular docking also provided useful insights into the development of new target molecules. The originality is good and the research content is comprehensive, and they also obtained some promising compounds. This manuscript can be considered for publication after minor revisions. The problems and suggestions are as follows:

1.        Why was the treatment concentration of the compound chosen as 50 μg·mL−1 in the antifungal activity testing, why don't you test at a lower concentration? So you can screen out more active compounds.

2.        The compounds data in the manuscript can be transferred into the Supporting Information.

3.        Can authors tell the difference between antifungal and fungicidal?

4.        The authors should revise the manuscript again, including the grammar, syntax and the explanation of data, make sure this manuscript is clear to the readers.

5.        The name of the tested plant pathogenic fungi should be written in Italic.

6.        Some English expressions or spelling errors existed, please check carefully. For example, in the conclusion,compound 4a and 5a could be regards as the novel and promising antifungal agentsshould be “be regarded as”.

7.        It is suggested that the authors should continue to determine the activity of related enzymes activity of highly active compounds in the subsequent studies.

Author Response

Manuscript ID: marinedrugs-2054963Title: Discovery of novel pimprinine and streptochlorin derivatives as potential antifungal agents  Dear Reviewer, 

Thanks a lot for your kind help, we also appreciate for the positive comments, patient checking and constructive suggestions to improve our manuscript, and this will also encourage us to carry out better research. Based on the comments and suggestions, we have revised our manuscript as described below.

The comments and suggestions: The manuscript entitled “Discovery of novel pimprinine and streptochlorin derivatives as potential antifungal agents” by Jing-Rui Liu et al. describes their latest study on the marine active compounds streptochlorin and its derivatives. As we know, developing new green pesticides is becoming increasingly difficult. Liu and co-authors used pimprinine and streptochlorin as the parent structures, designed and synthesized three series of new derivatives. The antifungal activity testing revealed some compounds showed excellent and broad-spectrum antifungal activity, several compounds were even more active than the most used fungicides. The SAR analysis and molecular docking also provided useful insights into the development of new target molecules. The originality is good and the research content is comprehensive, and they also obtained some promising compounds. This manuscript can be considered for publication after minor revisions. The problems and suggestions are as follows: 

1. Why was the treatment concentration of the compound chosen as 50 μg·mL−1 in the antifungal activity testing, why don't you test at a lower concentration? So you can screen out more active compounds. 

Answer: Thanks for your suggestion. We considered both the working effectiveness and the potential of the compounds when selecting the treatment concentration. If the concentration is too high, it will lead to generally good activity of all compounds, and it is difficult to screen compounds with really good activity. However, when the concentration is too low, some potential active lead compounds may be ignored. Therefore, we chose a relatively moderate concentration, which is 50 μg·mL−1

2. The compounds data in the manuscript can be transferred into the Supporting Information.

Answer: We appreciate for your useful comments. We have transferred the compounds data in the manuscript into the Supporting Information. 

3. Can authors tell the difference between antifungal and fungicidal?

Answer: We are thankful for your positive comments, and we are glad to tell the difference. Antifungal means to inhibit the growth of fungi, while fungicidal means to kill fungi. We hope to inhibit the growth of fungi through green and balanced methods rather than completely kill them. 

4. The authors should revise the manuscript again, including the grammar, syntax and the explanation of data, make sure this manuscript is clear to the readers.

Answer: We appreciate the reviewer for your kind help and patient checking, and we corrected all the grammar and syntax errors we had found. We also checked the explanation of data in the manuscript. 

5. The name of the tested plant pathogenic fungi should be written in Italic.

Answer: Thanks for your positive comments. The names of the tested plant pathogenic fungi have been written in Italic

6. Some English expressions or spelling errors existed, please check carefully. For example, in the conclusion, “compound 4a and 5a could be regards as the novel and promising antifungal agents” should be “be regarded as”.

Answer: We appreciate you for patient checking, all English expressions and spelling in the manuscript have been throughly checked and revised. 

7. It is suggested that the authors should continue to determine the activity of related enzymes activity of highly active compounds in the subsequent studies.

Answer: We appreciate the reviewer for the kind suggestions. Our plan is to finish the determination of related enzymes activity in the next study, we are now studying the relevant literatures.  

Thanks again for your kind help.

Sincerely yours,

Assoc. Prof. Ming-Zhi Zhang

Nanjing Agricultural University, Nanjing 210095, China

Tel: 86-25-8439-9210

Reviewer 2 Report

The article involves an interesting study to detect new antifungal compounds. The chemistry used to synthesize the compounds has been previously reported by other authors but provided an efficient way to acquire the compounds needed for this study.

The article was well written with just a few spelling errors which appear at the start of the article:

line 34 - should be: natural products not natural product

line 82 - should be: "addition time" not "feeding time"

The subsequent analysis of the compounds was interesting with the authors providing molecular modelling studies to account for the activity of the most active compounds.

Author Response

Manuscript ID: marinedrugs-2054963Title: Discovery of novel pimprinine and streptochlorin derivatives as potential antifungal agents  Dear Reviewer, 

Thanks a lot for your kind help, we also appreciate for the positive comments, patient checking and constructive suggestions to improve our manuscript, and this will also encourage us to carry out better research. Based on the comments and suggestions, we have revised our manuscript as described below.

The comments and suggestions: The article involves an interesting study to detect new antifungal compounds. The chemistry used to synthesize the compounds has been previously reported by other authors but provided an efficient way to acquire the compounds needed for this study.The article was well written with just a few spelling errors which appear at the start of the article:line 34 - should be: natural products not natural productline 82 - should be: "addition time" not "feeding time"The subsequent analysis of the compounds was interesting with the authors providing molecular modelling studies to account for the activity of the most active compounds.

Answer: We appreciate the reviewer for positive comments, which will motivate us to do better research, and also thanks for the patient checking, we corrected all the spelling errors we had found.

Thanks again for your kind help.

Sincerely yours,

Assoc. Prof. Ming-Zhi Zhang

Nanjing Agricultural University, Nanjing 210095, China

Tel: 86-25-8439-9210

Reviewer 3 Report

Manuscript ID: marinedrugs-2054963-peer-review-v1.pdf - “Discovery of novel pimprinine and streptochlorin derivatives as potential antifungal agents

The manuscript is very well written both in terms of chemical content and English spelling. The workflow adopted to achieve their goals are adequate, very well structured, presented and very actual.

Considering the potential impact of the manuscript results in the research world, and with all the respect for the author's work, the manuscript may be considered for publication in the Marine Drugs journal following minor revisions.  

In this regard, the authors are invited to make the following changes:

 1.       The authors prepared the ligands using the default settings of the Discovery Studio tool? is there a specific tool to prepare the ligands? Please give some details about. Also, please add in the Materials and Methods section some details about the molecular docking strategy.

2.       Did the authors use/identify more than one PDB structure?

3.   The bibliographic references are a bit old; it would be interesting and probably useful for some references to be updated. 

Author Response

Manuscript ID: marinedrugs-2054963Title: Discovery of novel pimprinine and streptochlorin derivatives as potential antifungal agents  Dear Reviewer, 

Thanks a lot for your kind help, we also appreciate for the positive comments, patient checking and constructive suggestions to improve our manuscript, and this will also encourage us to carry out better research. Based on the comments and suggestions, we have revised our manuscript as described below.

The comments and suggestions:Manuscript ID: marinedrugs-2054963-peer-review-v1.pdf - “Discovery of novel pimprinine and streptochlorin derivatives as potential antifungal agents” The manuscript is very well written both in terms of chemical content and English spelling. The workflow adopted to achieve their goals are adequate, very well structured, presented and very actual. Considering the potential impact of the manuscript results in the research world, and with all the respect for the author's work, the manuscript may be considered for publication in the Marine Drugs journal following minor revisions.  In this regard, the authors are invited to make the following changes:

1. The authors prepared the ligands using the default settings of the Discovery Studio tool? is there a specific tool to prepare the ligands? Please give some details about. Also, please add in the Materials and Methods section some details about the molecular docking strategy.

Answer: We appreciate the reviewer for positive comments and kind suggestions. We are glad to add some details. We prepared the protein and ligand using Discovery Studio. We deleted the water molecules in the protein, supplemented incomplete amino acid residues, and hydrotreated the protein. For ligand molecules, the software can be used to draw small molecules, optimize three-dimensional structure, conduct hydrotreatment and complete energy minimization. Of course, these processes can also be completed in Chemdraw and Chem3D together. But the final operation needs to be carried out in the Discovery Studio. Open the corresponding parameter setting panel, and generally set the default value. Then click Run to get the processed ligand molecule. We have added some details about the molecular docking strategy in the Materials and Methods section.

2.  Did the authors use/identify more than one PDB structure?

Answer: Thank the reviewer for the question. We used only one PDB structure due to our latest studies. We speculate that the target of this series of compounds may be similar to AN2690, so we take this protein as the target of molecular docking.

3. The bibliographic references are a bit old; it would be interesting and probably useful for some references to be updated. 

Answer: Thank the reviewer for useful suggestions. In this manuscript, we cited some original references that are related to the lead compounds and some classical methods’ references. Besides, we have updated some latest references.

Thanks again for your kind help.

Sincerely yours,

Assoc. Prof. Ming-Zhi Zhang

Nanjing Agricultural University, Nanjing 210095, China

Tel: 86-25-8439-9210

Reviewer 4 Report

Very interesting results Ming-Zhi Zhang and Yu-Cheng Gu group.

The iodine featuring indole oxazole synthesis is very interesting, it allows access to many derivatives in a fast manner (2-steps).

The study is detailed, well designed, and executed. Also, the report is clear and the compounds pure and well characterized.

In conclusion, a good manuscript, with novel chemistry, completed SAR study, molecular docking model study, non-trivial, with high levels of novelty and useful for the synthetic community and antifungal compound design.

I would recommend publication, essentially as it is.

I just have a quick question, why not compare the activity with the AN2690, Tavaborole, in the paper?

Author Response

Manuscript ID: marinedrugs-2054963Title: Discovery of novel pimprinine and streptochlorin derivatives as potential antifungal agents  Dear Reviewer, 

Thanks a lot for your kind help, we also appreciate for the positive comments, patient checking and constructive suggestions to improve our manuscript, and this will also encourage us to carry out better research. Based on the comments and suggestions, we have revised our manuscript as described below.

The comments and suggestions:Very interesting results Ming-Zhi Zhang and Yu-Cheng Gu group.The iodine featuring indole oxazole synthesis is very interesting, it allows access to many derivatives in a fast manner (2-steps).
The study is detailed, well designed, and executed. Also, the report is clear and the compounds pure and well characterized.
In conclusion, a good manuscript, with novel chemistry, completed SAR study, molecular docking model study, non-trivial, with high levels of novelty and useful for the synthetic community and antifungal compound design.
I would recommend publication, essentially as it is.I just have a quick question, why not compare the activity with the AN2690, Tavaborole, in the paper

Answer: Thanks for the reviewer's positive comments and his question. As described in the manuscript, the compounds we designed and synthesized are aiming to the pesticide applications, targeting plant pathogenic fungi, so we used commercial fungicides Osthole, Boscalid and Carbendazim as the positive controls, rather than AN2690, which is used to treat finger (toe) onychomycosis caused by Trichophyton rubrum and Trichophyton mentagrophyte, it belongs to the medical field.

Thanks again for your kind help.

Sincerely yours,

Assoc. Prof. Ming-Zhi Zhang

Nanjing Agricultural University, Nanjing 210095, China

Tel: 86-25-8439-9210
